# Comparative Study on Epidermal Moisturizing Effects and Hydration Mechanisms of Rice-Derived Glucosylceramides and Ceramides

**DOI:** 10.3390/ijms24010083

**Published:** 2022-12-21

**Authors:** Shogo Takeda, Akari Yoneda, Kenchi Miyasaka, Yoshiaki Manse, Toshio Morikawa, Hiroshi Shimoda

**Affiliations:** 1Research and Development Division, Oryza Oil and Fat Chemical Co., Ltd., 1 Numata, Kitagata-Cho, Ichinomiya 493-8001, Japan; 2Pharmaceutical Research and Technology Institute, Kindai University, 3-4-1 Kowakae, Higashi-Osaka 577-8502, Japan

**Keywords:** rice bran oil, glucosylceramide, elasticamide, transepidermal water loss, stratum corneum, filaggrin, corneodesmosin

## Abstract

Ceramide (Cer) plays an important role in skin barrier functions in the stratum corneum (SC). The ingestion of food-derived glucosylceramides (GlcCer) attenuates transepidermal water loss (TEWL). However, the moisturizing effects of single molecules of GlcCer and Cer remain unclear. Therefore, we herein purified 13 GlcCer and 6 Cer, including elasticamide, which has the same structure as human Cer[AP], from rice and compared their epidermal moisturizing effects in a reconstructed human epidermal keratinization model. The results obtained showed that 10 µM of 5 GlcCer[d18:2] with a 4*E*,8*Z* sphingadienine and C18 to C26 fatty acids and 10 µg/mL of 3 Cer with C23 or C24 fatty acids significantly reduced TEWL. The moisturizing effects of these GlcCer were dependent on the length of fatty acids. Furthermore, 10 µg/mL of elasticamide increased the SC Cer contents by promoting the expression of GlcCer synthase. Electron microscopic observations revealed that 1 µM of GlcCer[d18:2(4*E*,8*Z*)/26:0] increased the number of keratohyalin granules and desmosomes. Immunostaining and Western blotting indicated that 1 µM of GlcCer[d18:2(4*E*,8*Z*)/26:0] up-regulated the expression of filaggrin and corneodesmosin, which contribute to epidermal hydration. This comparative study on epidermal moisturization by GlcCer and Cer isolated from rice revealed differences in their hydration mechanisms.

## 1. Introduction

The epidermis is the outermost layer of the skin and contributes to skin barrier functions and moisturization. It consists of the following four sublayers in order from the inner- to outermost layers: the stratum basal, stratum spinosum, stratum granulosum, and stratum corneum (SC). Approximately 95% of cells in the epidermis are keratinocytes. SC is formed by keratinization and performs barrier functions, such as inhibiting water evaporation and protecting against foreign substances and ultraviolet light [1,2,3]. The barrier functions of SC are maintained by the lamellar structure, which is formed by intercellular lipids and water molecules and fills the spaces between SC cells. Intercellular lipids are mainly composed of ceramide (Cer), which accounts for approximately 50%, and cholesterol, fatty acids, and cholesterol sulfates [4]. However, defective epidermal barrier functions cause skin dehydration due to poor water retention and the skin surface becomes more sensitive to non-specific stimuli, such as protein antigens and ultraviolet light. Defects also result in the release of various cytokines from inflammatory cells, leading to the development and worsening of dermatitis and itching. Therefore, it is important to retain the factors that contribute to barrier functions in order to maintain a healthy epidermis.

Cer is characterized as a sphingolipid that consists of a sphingoid base and fatty acid moiety. It is a component of the mammal cell membrane, a dominant lipid in the SC of human skin, and plays important roles in moisturization and barrier functions [5]. In a recent profiling study on Cer species in the human SC, 1327 species of Cer, including 20 major classes, were identified [6]. SC Cer is synthesized by several sequential enzymatic reactions involving the following enzymes. Serine palmitoyltransferase (SPT) and ceramide synthase (CerS) play roles in the de novo synthesis of Cer. SPT catalyzes the condensation of serine and palmitoyl-CoA as the first step of de novo synthesis [7], while CerS catalyzes the formation of the basic Cer structure through the *N*-acyltranslation of fatty acids [8]. Cer in SC is synthesized from glucosylceramides (GlcCer) and sphingomyelin, and GlcCer synthase (GCS), β-glucocerebrosidase (GBA), sphingomyelin synthase (SMS), and acid sphingomyelinase (ASM) are involved in the synthesis and metabolism of these Cer precursors. GCS plays a role in GlcCer synthesis [9] and GlcCer are hydrolyzed to Cer by GBA [10]. SMS catalyzes the synthesis of sphingomyelin [11], while ASM synthesizes Cer from sphingomyelin [12]. Previous studies reported that SC Cer content was significantly decreased in patients with atopic dermatitis (AD) [13] and xerosis [14] and decreases in Cer correlated with increased transepidermal water loss (TEWL) [15]. Therefore, the maintenance of or increases in SC Cer content is one of the approaches that effectively suppresses epidermal dehydration.

GlcCer are found in plant materials as a major sphingoid lipid and are widely distributed in various plants [16], while they are the precursor of SC Cer in human skin. Previous studies demonstrated that dietary plant-derived GlcCer enhanced skin barrier functions. The oral administration of GlcCer derived from rice [17,18], maize [18], konjac [19], and pineapples [20] has been reported to attenuate TEWL in hairless mice. Furthermore, human clinical tests demonstrated the epidermal hydration effects of dietary plant-derived GlcCer [19,21,22,23,24,25,26]. We also showed that the ingestion of rice-derived GlcCer improved facial conditions, including facial TEWL, lip moisture, and the size of visible pores [27]. These findings indicate that plant-derived GlcCer have the potential to improve skin barrier functions and epidermal hydration. However, it is important to note that these studies employed a crude GlcCer fraction as the test substance.

As described above, the effects of single molecules of GlcCer, distinguished by the type of fatty acid and sphingoid base, currently remain unclear. We recently isolated 13 species of GlcCer and 6 of Cer, including elasticamide, which has the same structure as human Cer[AP], from rice and examined the effects of these substances on melanogenesis [28] (Figure 1 and Figure 2). In the present study, we compared the epidermal moisturizing effects of isolated GlcCer and Cer in a human epidermal equivalent model. Based on the results obtained, the potential hydration mechanisms of GlcCer and Cer can be summarized and discussed.

## 2. Results

### 2.1. Effects of GlcCer (***1***–***13***) and Cer (***14***–***19***) on TEWL in the Reconstructed Human Epidermal Keratinization (RHEK) Model

We examined the effects of GlcCer (**1**–**13**) and Cer (**14**–**19**) on TEWL in the RHEK model to compare their moisturizing activities. The results obtained showed that 10 µM of GlcCer, which has a [d18:2(4*E*,8*Z*)] structure in the sphingoid base, including GlcCer[d18:2(4*E*,8*Z*)/18:0] (**1**), GlcCer[d18:2(4*E*,8*Z*)/20:0] (**3**), GlcCer[d18:2(4*E*,8*Z*)/22:0] (**7**), GlcCer[d18:2(4*E*,8*Z*)/24:0] (**10**), and GlcCer[d18:2(4*E*,8*Z*)/26:0] (**13**), significantly decreased TEWL after 7 days of treatment (Figure 3). The effects of these GlcCer were dependent on the length of fatty acids, with **13** exerting the strongest moisturizing effects. Regarding the Cer treatment, 10 µg/mL of Cer[t18:0/24:0] (**14**, elasticamide), Cer[t18:0/23:0] (**16**), and Cer[t18:1(8*Z*)/24:0] (**18**) significantly reduced TEWL, and the moisturizing effects of **14** were as strong as those of **13**. Furthermore, **14** exhibited concentration-dependent activity at 1, 3, and 10 µg/mL (Figure 3, bottom).

### 2.2. Effects of GlcCer (***1***–***13***) and Cer (***14***–***19***) on SC Cer contents in the RHEK Model

To elucidate hydration mechanisms, we examined the effects of GlcCer (**1**–**13**) and Cer (**14**–**19**) on SC Cer contents. Regarding GlcCer, SC Cer contents were not affected by any of the tested molecules at 10 µM (Figure 4A). On the other hand, **14** and **18**, which attenuated TEWL in the RHEK model, significantly increased SC Cer contents (Figure 4B). Therefore, we confirmed the concentration dependency of **14**, which showed the strongest activity on TEWL improvement among the Cer examined. Additionally, **14** increased SC Cer contents in a concentration-dependent manner, with 10 µg/mL of **14** significantly increasing total Cer and Cer[NS,NDS] (Figure 5).

### 2.3. Effects of Elasticamide (***14***) on the Expression of Enzymes Related to SC Cer Synthesis in the RHEK Model

Elasticamide (**14**) significantly increased SC Cer contents in the RHEK model, as described above. Therefore, we examined the effects of **14** on the expression of enzymes involved in SC Cer synthesis. Figure 6A shows the effects of **14** on the mRNA expression of enzymes related to SC ceramide synthesis in the RHEK model: SPT2, CerS3, GCS, GBA, SMS2, and ASM. Furthermore, **14** significantly up-regulated the mRNA expression of GCS and SMS2 at 1 and 10 µg/mL, respectively. Figure 6B shows the effects of **14** on the protein expression of these enzymes; it significantly up-regulated the protein expression of GCS at 1 and 3 µg/mL, but not that of SMS2.

### 2.4. Effects of GlcCer[d18:2(4E,8Z)/26:0] (***13***) on Epidermal Hydration Factors in the RHEK Model

As described above, GlcCer did not affect SC Cer contents, while elasticamide (**14**) increased SC Cer via the up-regulation of GCS expression. Therefore, we conducted further experiments to elucidate the epidermal hydration mechanism of **13**, which exerted the strongest moisturizing effects. The concentration-dependent effects of **13** (1, 3, and 10 µM) on TEWL were confirmed (Figure 7A). However, similar to the results shown in Figure 4, none of the concentrations of **13** increased SC Cer contents (Figure 7B). Accordingly, we performed electron microscopic observations of cross-sections of the RHEK model. The results obtained indicated that 1 µM of **13** increased the intercellular density of the SC layer. Furthermore, **13** appeared to increase the number of keratohyalin granules and desmosomes (Figure 7C).

### 2.5. Effects of GlcCer[d18:2(4E,8Z)/26:0] (***13***) on the Protein Expression of Filaggrin and Corneodesmosin

Since electron microscopic observations indicated that **13** increased keratohyalin granules and desmosomes, we examined the effects of **13** on the protein expression of filaggrin and corneodesmosin. Figure 8A shows immunostaining and Western blotting images of filaggrin expression. Based on immunostaining images, **13** (1 µM) appeared to up-regulate the expression of filaggrin in SC and Western blotting showed significant increases in the protein expression of filaggrin by **13** (1 and 3 µM). As shown in Figure 8B, immunostaining images suggested the up-regulated expression of corneodesmosin in SC by **13** (1 µM) and Western blotting revealed that **13** (1 µM) significantly increased the protein expression of corneodesmosin.

## 3. Discussion

In the present study, we isolated 13 GlcCer and 6 Cer from by-products of rice bran oil and compared their effects on epidermal moisturization in a RHEK model. We initially investigated the effects of GlcCer (**1**–**13**) and Cer (**14**–**19**) on TEWL. TEWL is the most widely used measurement to evaluate epidermal barrier functions [29]. Significant increases in TEWL are commonly observed in dry skin diseases, such as AD [30], xerosis, and psoriasis [31]. We previously reported the moisturizing effects of isolated steroidal saponins from tomato seeds and β-sitosterol 3-*O*-glucoside from rice on TEWL in RHEK models [32]. In the present study, all GlcCer[d18:2(4*E*,8*Z*)] (**1**, **3**, **7**, **10**, and **13**) significantly attenuated TEWL, whereas GlcCer[t18:1(8*Z*)] (**2**, **5**, **9**, and **12**), GlcCer[d18:2(4*E*,8*E*)] (**4**, **8** and **11**), and GlcCer[t18:1(4*E*)] (**6**) did not. These results suggest that the d18:2(4*E*,8*Z*) structure is required for the moisturizing effects of GlcCer. Furthermore, reductions in TEWL were dependent on the length of fatty acids, and the moisturizing effects of **10** and **13**, which have C24 or more fatty acids, were detected in the earlier stages of the culture (Days 1 and 3). Therefore, a longer fatty acid length appeared to accelerate the epidermal hydration activity of GlcCer. Previous studies demonstrated the effects of plant-derived GlcCer on TEWL. Tsuji et al. reported that the oral administration of GlcCer from rice or maize attenuated TEWL in hairless mice [18], while Kuwata et al. showed that dietary GlcCer from pineapples reduced TEWL in hairless mice [20]. Our previous findings revealed that the oral administration of rice-derived GlcCer reduced TEWL in the skin of SDS-treated hairless mice [17]. However, these experiments used a mixture of GlcCer as the test substance and a comparison of the effects of single molecules of GlcCer has not yet been conducted. Therefore, the present study is the first report to compare and evaluate the effects of single molecules of GlcCer on epidermal hydration. Regarding Cer species, the significant attenuation of TEWL was observed with Cer[t18:0/24:0] (**14**, elasticamide), Cer[t18:0/23:0] (**16**), and Cer[t18:1(8*Z*)/24:0] (**18**). However, in contrast to the activities of GlcCer (**10**, **13**), Cer[t18:0/25:0] (**17**) and Cer[t18:0/26:0] (**19**), which have more than C24 fatty acids, did not affect TEWL. Moreover, although none of the tested GlcCer[t18:1(8*Z*)] (**2**, **5**, **9**, and **12**) affected TEWL, Cer (**18**) with the t18:1(8*Z*) structure significantly attenuated TEWL. On the other hand, Cer (**14**) and GlcCer (**13**) both reduced TEWL in the later stages of the culture (Days 5 and 7), while only GlcCer (**13**) attenuated TEWL from the earlier stage (Day 1). Collectively, these results revealed differences in the hydration mechanisms of GlcCer and Cer.

To elucidate the hydration mechanisms of GlcCer and Cer, we quantified SC Cer contents in the RHEK model treated with GlcCer (**1**–**13**) and Cer (**14**–**19**). In a profiling study on Cer species, 20 major classes of Cer were identified in the human SC [6]. SC Cer contents significantly decreased in patients with AD [13] and xerosis [14], and reductions in SC Cer contents resulted in epidermal water evaporation [15]. In the present study, 10 µg/mL of **14** and **18** significantly increased total SC Cer contents in the RHEK model. Moreover, **14** increased SC Cer contents in a concentration-dependent manner in concert with a significant increase in Cer[NS,NDS], which is the most abundant Cer species in the SC. Although the structure of **14** is the same as that of human Cer[AP], **14** did not increase SC Cer[AP] contents. This result suggests that the epidermal hydration mechanism of **14** does not involve a direct supply of Cer to the SC from the medium under the RHEK membrane. Therefore, we examined the effects of **14** on the mRNA and protein expression of SPT2, CerS3, GCS, GBA, SMS2, and ASM, which are involved in SC ceramide synthesis [7,8,9,10,11,12]. The results obtained showed that **14** significantly up-regulated the expression of GCS, which acts as a glucosyltransferase enzyme to synthesize GlcCer, the precursors of SC Cer [9]. Therefore, **14** increased SC Cer contents by up-regulating the expression of GCS, leading to reductions in TEWL. We demonstrated that tiliroside, the main component of strawberry seed extract, increased SC Cer[NS,NDS] by up-regulating GCS and GBA [33]. Although **14** did not significantly up-regulate the expression of GBA, its protein expression level was slightly increased. Consequently, the mechanisms by which tiliroside and **14** increase SC Cer contents are considered to be similar.

Regarding the hydration mechanism of rice GlcCer, **1**, **3**, **7**, **10**, and **13** did not affect SC Cer contents even though they significantly attenuated TEWL. Furthermore, between 1 and 10 µM of **13** did not increase SC Cer contents but dose-dependently reduced TEWL. Accordingly, the epidermal hydration mechanism of **13** differed from that of **14**. Electron microscopic observations to investigate the hydration mechanism of **13** revealed that the intercellular density of SC layers was increased by **13**, indicating better moisture retention. In addition, the number of keratohyalin granules appeared to be increased by **13**. Keratohyalin granules are present in the stratum granulosum and consist of profilaggrin, which is a precursor of filaggrin. Although profilaggrin does not bind to keratin, it is dephosphorylated and proteolyzed into filaggrin monomers during the later stages of the keratinization process [34]. Filaggrin then provides natural moisturizing amino acids through its decomposition in the SC [35] and acts as a filament-aggregating protein [36]. Previous studies reported that the number of keratohyalin granules and filaggrin expression were reduced or absent in the skin of patients with ichthyosis vulgaris; therefore, the number of keratohyalin granules in the stratum granulosum reflect filaggrin production levels [37,38]. Herein, we examined the effects of **13** on filaggrin protein expression in the RHEK model. Immunostaining observations and Western blotting results revealed that the protein expression of filaggrin was significantly increased by **13**. These results suggest that the increase in filaggrin, which functions as a natural moisturizing factor in the SC, may be the epidermal hydration mechanism of **13**.

In addition, electron microscopic observations suggested an increase in the number of desmosomes by **13**. Desmosomes are intercellular junctions that provide strong adhesion between epidermal cells [39,40] and their shape markedly changes from corneodesmosomes in the SC [41]. Corneodesmosin, which is specifically found in corneodesmosomes, plays a pivotal role in SC cell adhesion [42]. Furthermore, an analysis of corneodesmosin-knockout mouse skin revealed the detachment of the SC from the living layers of the epidermis, which resulted in prominent defects in barrier functions [43]. Therefore, increases in the expression of corneodesmosin are considered to be a reasonable mechanism for epidermal hydration. Immunostaining observations and Western blotting results revealed that the protein expression of corneodesmosin was significantly up-regulated by **13**. Therefore, changes in corneodesmosin may be another hydration mechanism of **13** that reduces TEWL.

## 4. Conclusions

Herein, we demonstrated that among the 13 GlcCer isolated from rice, d18:2(4*E*,8*Z*)-type GlcCer reduced TEWL in a fatty acid length-dependent manner. The hydration mechanism of GlcCer appears to involve increases in SC density and the expression of filaggrin and corneodesmosin. Regarding the 6 isolated Cer, Cer with C23 or C24 fatty acids also reduced TEWL. Moreover, elasticamide (**14**) has been suggested to increase SC Cer contents (particularly Cer[NS,NDS]) via the up-regulated expression of GCS. The present study is the first to compare the epidermal moisturizing effects of GlcCer and Cer species as single compounds and reveal differences in the hydration mechanisms of GlcCer and Cer.

## 5. Materials and Methods

### 5.1. Preparation of GlcCer and Cer

To isolate GlcCer and Cer, they were purified according to our previously described methods [27,28]. The gummy by-products obtained during the manufacturing process of rice bran oil were subjected to flash column chromatography equipped with a silica gel column and evaporative liquid scattered detector (ELSD) to obtain crude GlcCer and Cer fractions. The crude GlcCer fraction was separated and repeatedly purified by HPLC to obtain GlcCer[d18:2(4*E*,8*Z*)/18:0] (**1**), GlcCer[t18:1(8*Z*)/20:0] (**2**), GlcCer[d18:2(4*E*,8*Z*)/20:0] (**3**), GlcCer[d18:2(4*E*,8*E*)/20:0] (**4**), GlcCer[t18:1(8*Z*)/22:0] (**5**), GlcCer[d18:1(4*E*)/20:0] (**6**), GlcCer[d18:2(4*E*,8*Z*)/22:0] (**7**), GlcCer[d18:2(4*E*,8*E*)/22:0] (**8**), GlcCer[t18:1(8*Z*)/24:0] (**9**), GlcCer[d18:2(4*E*,8*Z*)/24:0] (**10**), GlcCer[d18:2(4*E*,8*E*)/24:0] (**11**), GlcCer[t18:1(8*Z*)/26:0] (**12**), and GlcCer[d18:2(4*E*,8*Z*)/26:0] (**13**). The isolated yields from **1**–**13** were described in our previous study [28].

To obtain each Cer, the crude Cer fraction was purified by HPLC equipped with an ODS column (Capcell pak C18 SG-120, ϕ 20 × 250 mm, Osaka Soda, Osaka, Japan) and the RI detector, MeOH: tetrahydrofuran (9:1) as the mobile phase to obtain Cer[t18:0/24:0] (**14**, elasticamide, 9.9 mg), Cer[t18:0/22:0] (**15**, 6.2 mg), Cer[t18:0/23:0] (**16**, 7.4 mg), Cer[t18:0/25:0] (**17**, 4.6 mg), Cer[t18:1(8*Z*)/24:0] (**18**, 5.1 mg), and Cer[t18:0/26:0] (**19**, 5.5 mg). Chemical structures were identified by ^1^H- and ^13^C- NMR spectra with the referenced values listed in previous studies [44,45,46,47,48,49].

### 5.2. Reagents

Trypsin (0.25 *w/v*%)/EDTA (1 mmol/L)·4Na solution with phenol red (trypsin/EDTA solution), phosphate-buffered saline (PBS), L.A.B solution, 3,3′-diaminobenzidine (DAB) tablets, Mayer’s Hematoxylin Solution, and skim milk were obtained from FUJIFILM Wako Pure Chemical Co., Ltd. (Osaka, Japan). The RNeasy^®^ Mini Kit was purchased from QIAGEN (Hilden, Germany). PrimeScriptTM Reverse Transcriptase and TB Green^®^ Premix Dimer EraserTM were purchased from Takara Bio Inc. (Kusatsu, Japan). The dNTP mixture, random primers, and anti-glyceraldehyde 3-phosphate dehydrogenase (GAPDH) antibody (GA1R) were purchased from Invitrogen (Carlsbad, CA, USA). The high performance thin-layer chromatography (HPTLC) plate, horseradish peroxidase (HRP)-conjugated goat anti-rabbit IgG and HRP-conjugated goat anti-mouse IgG were purchased from Merck Millipore (Darmstadt, Germany). The standard compounds of Cer[NS,NDS] and [AS] were purchased from Matreya LLC (State College, PA, USA). Radioimmunoprecipitation assay lysis buffer, the protease phosphatase inhibitor cocktail, BCA protein assay kit, SuperSignal™ West Pico PLUS Chemiluminescent Substrate, and SuperSignal™ West Femto Maximum Sensitivity Substrate were purchased from Thermo Fisher Scientific Inc. (Waltham, MA, USA). Anti-SPT2 (ab23696), anti-CerS3 (ab28637), anti-ASM (ab83354), anti-filaggrin (ab17808), and anti-corneodesmosin (ab204235) antibodies were obtained from Abcam, Inc. (Cambridge, UK). Anti-GCS (sc-50511) and anti-SMS2 (sc-34048) antibodies were purchased from Santa Cruz Biotechnology, Inc. (Littleton, CO, USA). An anti-GBA antibody (G4171) was supplied by Sigma-Aldrich Co., LLC (St. Louis, MO, USA).

### 5.3. Culture of a Reconstructed Human Epidermal Keratinization (RHEK) Model

A RHEK model (LabCyte EPI-MODEL) and assay medium obtained from Japan Tissue Engineering Co., Ltd. (Gamagori, Japan) were used in the experiments. Each cup of the RHEK models were placed into a 24-well culture plate and assay medium was added underneath the cup. After incubating the plate at 37 °C under a 5% CO_2_ atmosphere for 1 day, the RHEK model was treated with solutions of GlcCer (**1**–**13**) or Cer (**14**–**19**) (final DMSO concentration: 0.1%). Culture times were selected for each experiment. The RHEK model was cultured for 7 days and then subjected to TEWL measurements, a lipid analysis, Western blotting, and microscopic observations, or for 2 days followed by real-time RT-PCR.

### 5.4. Measurement of TEWL in the RHEK Model

TEWL measurements were performed according to previously described methods [32,50,51]. Measurements were conducted before and 1, 3, 5, and 7 days after the treatment of samples with Tewitro TW24 (Courage + Khazaka, Cologne, Germany), which is the device specifically developed for the TEWL measurement in vitro testing and can directly and simultaneously measure the TEWL of the RHEK model in a 24-well plate. The RHEK model was placed on a thermal insulation mat (HIENAI Mat; Cosmo Bio Co., Ltd., Tokyo, Japan), which consistently kept the entire bottom surface at 32 °C, without a lid for 5 min before measurements. TEWL was measured for 30 min by inserting the sensor of Tewitro TW24 into each well of the RHEK model while maintaining the bottom surface at 32 °C, and the mean value for the last 10 min was used in the analysis. Measurements were performed under sterile conditions.

### 5.5. Lipid Extraction and Cer Determination

Lipid extraction and Cer determination were conducted according to our previously described methods [32,33,52]. SC lipids from the RHEK model were extracted using a mixture of CHCl_3_, MeOH, and PBS (1:2:0.8). Dried SC lipid samples were subjected to HPTLC to measure Cer contents.

### 5.6. Real Time RT-PCR

The mRNA expression of Cer synthesis-related enzymes in the RHEK model was measured by quantitative real-time RT-PCR according to our previously described method [32]. Extracted RNA from the whole RHEK model was reverse-transcribed to obtain cDNA. Information on specific primers was previously described [32].

### 5.7. Western Blotting Analysis

The Western blotting analysis described in our previous studies [33,52] was used to assess protein expression. Proteins extracted from the RHEK model were electrophoresed on 10% SDS gels and separated proteins were transferred to polyvinylidene difluoride membranes. After blocking with 5% skim milk, membranes were treated with a primary antibody followed by a secondary antibody. Anti-SPT2 (1:1000), anti-GCS (1:200), anti-CerS3 (1:1000), anti-GBA (1:1000), anti-SMS2 (1:150), anti-ASM (1:150), anti-filaggrin (1:200), anti-corneodesmosin (1:200), and anti-GAPDH (1:1000) antibodies were used as primary antibodies. HRP-conjugated goat anti-rabbit IgG (1:5000) and HRP-conjugated goat anti-mouse IgG (1:5000) were used as secondary antibodies. SuperSignal™ West Pico PLUS Chemiluminescent Substrate and SuperSignal™ West Femto Maximum Sensitivity Substrate were used for detection solutions.

### 5.8. Microscopic Analysis and Immunostaining

In immunohistochemistry of the RHEK model, formalin/paraformaldehyde-fixed tissues were embedded in paraffin to obtain specimens at Aichi Pathology Laboratory LLC (Mihama, Japan). Deparaffinized specimens were treated with 0.3% H_2_O_2_ in MeOH and soaked in L.A.B solution for 15 min to activate antigens. After blocking with 5% skim milk for 1 h, specimens were treated with the primary antibody followed by the secondary antibody. Anti-filaggrin (1:200) and anti-corneodesmosin (1:200) antibodies were used as the primary antibodies. HRP-conjugated goat anti-rabbit IgG (1:5000) was used as the secondary antibody. Specimens were then treated with DAB solution followed by staining with hematoxylin solution. Microscopic images were taken by a Leica DMI6000 (Leica Microsystems GmbH, Wetzlar, Germany).

Regarding electron microscopic observations of tissues, the preparation and imaging of pathological sections were consigned to the Hanaichi UltraStructure Research Institute, Co., Ltd. (Okazaki, Japan). The samples of tissues for TEM were fixed in phosphate buffered 2% glutaraldehyde, and subsequently post-fixed in 2% osmium tetra-oxide for 2 h in an ice bath. Then, the specimens were dehydrated in a graded ethanol and embedded in the epoxy resin. Ultrathin sections were obtained by ultramicrotome technique. Ultrathin sections stained with uranyl acetate for 15 min and lead staining solution for 2 min were submitted to TEM observation at 100 kV (H-7600, HITACHI Ltd., Tokyo, Japan).

### 5.9. Statistical Analysis

All statistical analyses were performed using Statcel (OMS Ltd., Tokyo, Japan) as an add-in to excel. All results are expressed as means ± standard errors (S.E.). The significance of differences was examined by a one-way analysis of variance followed by Dunnett’s test, with *p* < 0.05 or *p* < 0.01 indicating a significant difference.

## Figures and Tables

**Figure 1 ijms-24-00083-f001:**
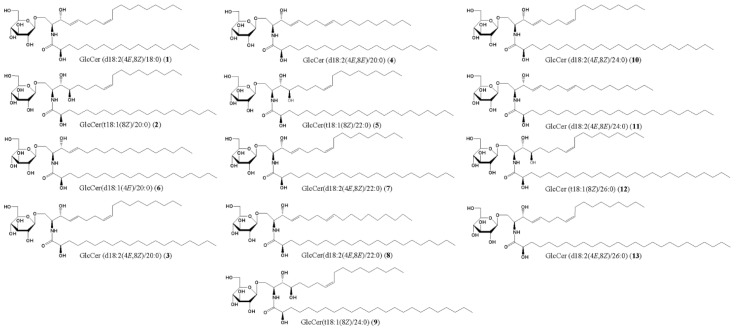
Chemical structures of GlcCer (**1**–**13**) isolated from GlcCer-rich rice oil by-product.

**Figure 2 ijms-24-00083-f002:**
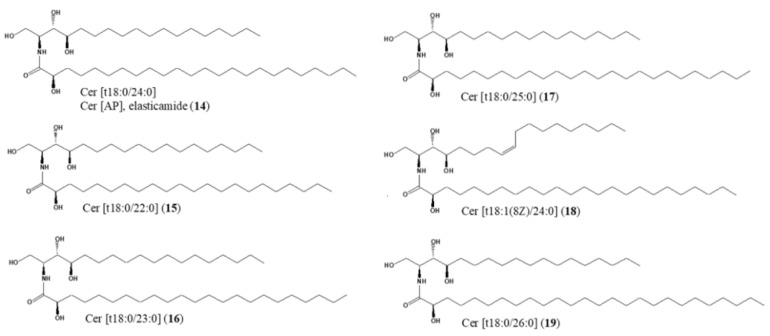
Chemical structures of Cer (**14**–**19**) isolated from GlcCer-rich rice oil by-product.

**Figure 3 ijms-24-00083-f003:**
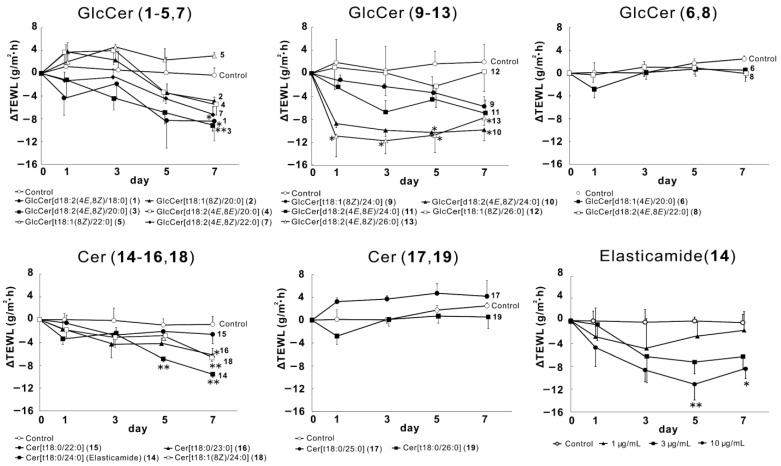
Effects of GlcCer (**1**–**13**) and Cer (**14**–**19**) on TEWL in the RHEK model. The RHEK model was treated for 7 days with each sample (**1**–**13**: 10 µM, **14**: 1–10 µg/mL, **15**–**19**: 10 µg/mL). TEWL measurements were performed on days 0, 1, 3, 5, and 7 using Tewitro TW24. Data are expressed as means ± S.E. (n = 3–4). Asterisks denote a significant difference from the control group, * *p* < 0.05, ** *p* < 0.01.

**Figure 4 ijms-24-00083-f004:**
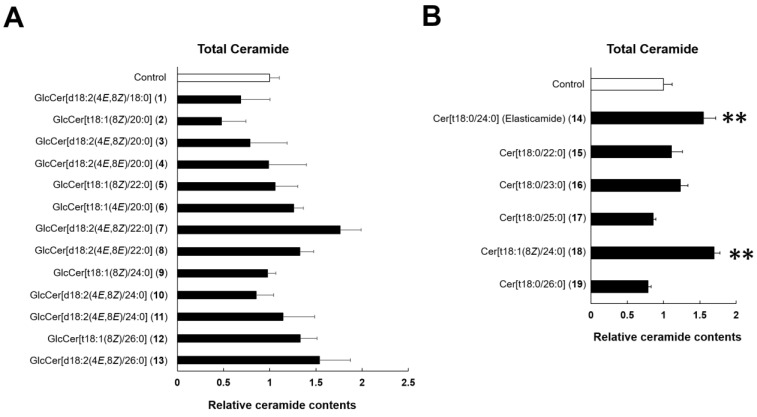
Effects of GlcCer (**1**–**13**) and Cer (**14**–**19**) on SC ceramide contents in RHEK models. (**A**) Effects of GlcCer (**1**–**13**); (**B**) Effects of Cer (**14**–**19**). The RHEK model was treated for 7 days with GlcCer (**1**–**13**: 10 µM) or Cer (**14**–**19**: 10 µg/mL). The extraction of lipids from the SC of the RHEK model and a high performance thin-layer chromatography (HPTLC) analysis were performed as described in the Materials and Methods section. The effects of each GlcCer are corrected for the control value. Data are expressed as means ± S.E. (n = 3–4). Asterisks denote a significant difference from the control group, ** *p* < 0.01.

**Figure 5 ijms-24-00083-f005:**
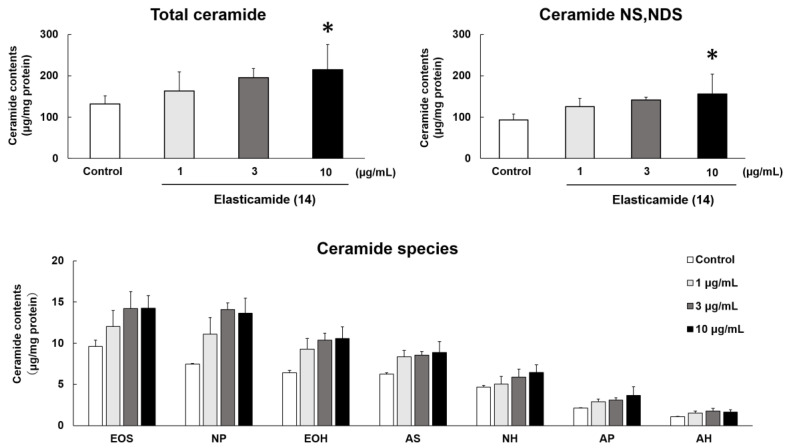
Effects of elasticamide (**14**) on SC ceramide contents in the RHEK model. The RHEK model was treated for 7 days with **14** (1, 3, and 10 µg/mL). The extraction of lipids from the SC of the RHEK model and a HPTLC analysis were performed as described in the Materials and Methods section. Data are expressed as means ± S.E. (n = 3–4). Asterisks denote a significant difference from the control group, * *p* < 0.05.

**Figure 6 ijms-24-00083-f006:**
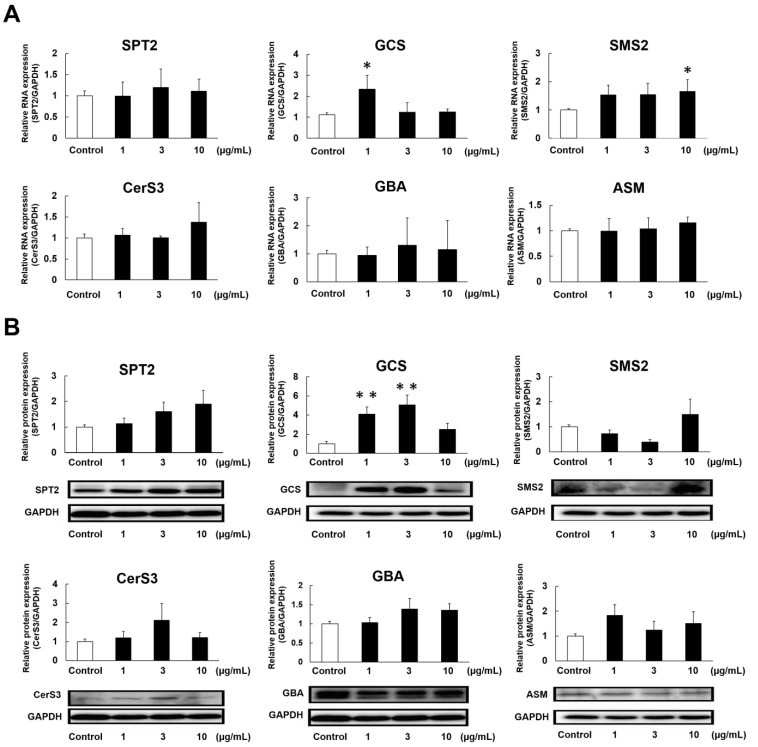
Effects of elasticamide (**14**) on the mRNA (**A**) and protein (**B**) expression of enzymes related to SC Cer synthesis. The RHEK model was treated for 2 days (for real-time RT-PCR) or 7 days (for Western blotting) with **14** (1–10 µg/mL). Real-time RT-PCR and a Western blotting analysis were performed as described in the Materials and Methods section. Data are expressed as means ± S.E. (n = 4). Asterisks denote a significant difference from the control group, * *p* < 0.05, ** *p* < 0.01.

**Figure 7 ijms-24-00083-f007:**
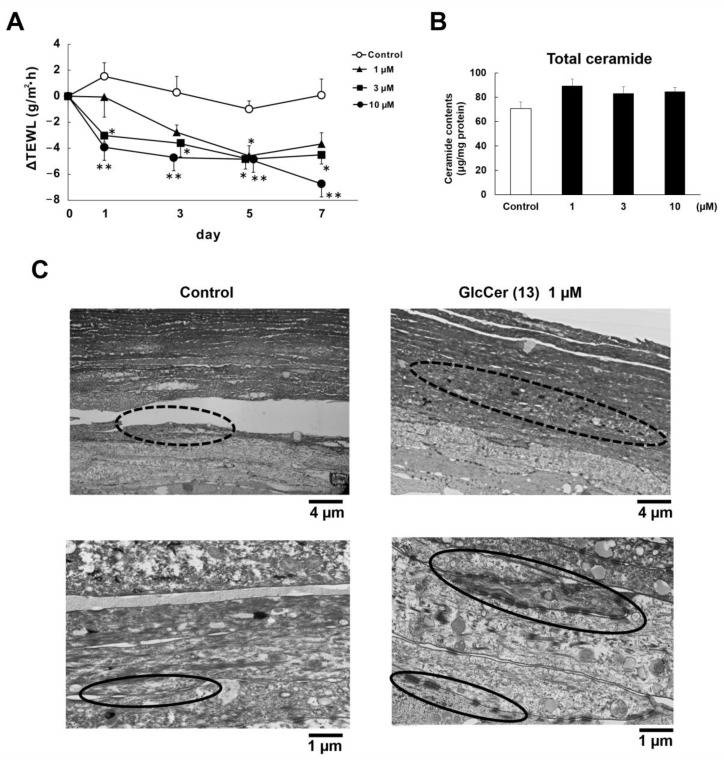
Effects of GlcCer[d18:2(4*E*,8*Z*)/26:0)] (**13**) on epidermal hydration factors in the RHEK model. (**A**) Changes in TEWL (n = 6); (**B**) SC Cer contents after the treatment with **13** for 7 days (n = 4); (**C**) Electron microscopic analysis of cross-sections of the RHEK model. Dotted and solid circles indicate keratohyalin granules and desmosomes, respectively. All data are expressed as means ± S.E. Asterisks denote a significant difference from the control group, * *p* < 0.05, ** *p* < 0.01.

**Figure 8 ijms-24-00083-f008:**
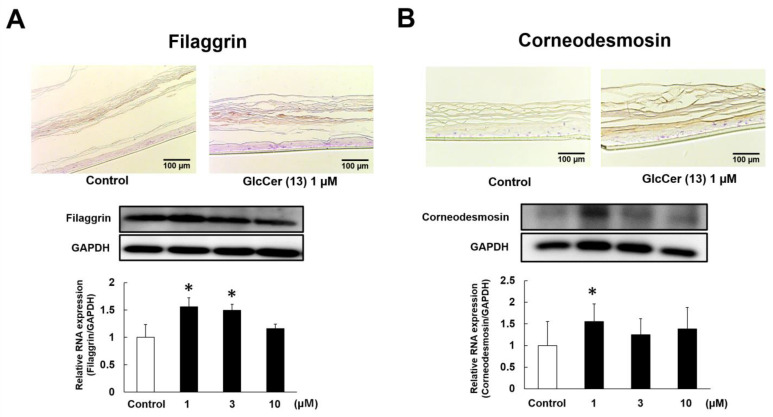
Effects of GlcCer[d18:2(4*E*,8*Z*)/26:0)] (**13**) on the protein expression of filaggrin (**A**) and corneodesmosin (**B**). The RHEK model was treated for 7 days with **13**. Immunostaining and Western blotting were performed as described in the Materials and Methods section. Data are expressed as means ± S.E. (n = 4–5). Asterisks denote a significant difference from the control group, * *p* < 0.05.

## Data Availability

The data that support the findings of this study are available from the corresponding author upon reasonable request.

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
