# Peer review of "Comparative Study on Epidermal Moisturizing Effects and Hydration Mechanisms of Rice-Derived Glucosylceramides and Ceramides"

_ijms, 2022, doi:10.3390/ijms24010083_

Round 1
Reviewer 1 Report
ijms-2090403: Shogo Takeda, Akari Yoneda, Kenchi Miyasaka, Yoshiaki Manse, Toshio Morikawa, Hiroshi Shimoda, “Comparative Study on Epidermal Moisturizing Effects and Hydration Mechanisms of Rice-derived Glucosylceramides and Ceramides”
The authors performed a detailed experimental comparative study of the epidermal moisturizing effects of rice-derived glucosylceramides and ceramide species. The research topic is of high scientific interest and the results are well presented. However, I suggest the addition of a new section 4, to be named: Conclusions (or Conclusion), with the text similar to what is written on lines 267-275. The recommended first sentence to type in is: “We herein demonstrated that…”
Author Response
Response:
We have added a new section 4 as a “Conclusion” section. The descriptions are as follows (Page 10, Lines 267-276).
“We herein demonstrated that among the 13 GlcCer isolated from rice, d18:2(4E,8Z)-type GlcCer reduced TEWL in a fatty acid length-dependent manner. The hydration mechanism of GlcCer appears to involve increases in SC density and the expression of filaggrin and corneodesmosin. Regarding the 6 isolated Cer, Cer with C23 or C24 fatty acids also reduced TEWL. Moreover, elasticamide (14) has been suggested to increase SC Cer contents (particularly Cer[NS,NDS]) via the up-regulated expression of GCS. The present study is the first to compare the epidermal moisturizing effects of GlcCer and Cer species as single compounds and reveal differences in the hydration mechanisms of GlcCer and Cer.”
Reviewer 2 Report
Minor English edition is required.
Author Response
Response:
We have confirmed descriptions in manuscript and amended some points of concern as follows.
Page 1, Lines 38-40.
Page 4, Line 111.
Page 4, Lines 113-114.
Page 8, Line 202.
Reviewer 3 Report
The article describes a comparative study of the moisturising effect of different ceramides, as it is known that glucosylceramides attenuate transepidermal water loss. For this, the authors have utilised a wide number of techniques, including cell culture, RT-PCR, Western blot and microscopy. The figures are well done, although Figure 3 needs to be improved. The methods need to be improved, please see comments. I’m suggesting the authors to check on a few things before accepting this article for publication.
· Figure 3: please improve quality. The lines look blurred and the legend in the graph at the bottom left has been cut.
· Section 4.4: could the authors describe the method? The authors mentioned they treated the samples with Tewitro, what is Tewitro?
· Section 4.7: please describe in depth electron microscopy method. SEM or TEM? Which microscope was used? Which parameters were used (working distance, accelerating voltage)? How where the samples prepared?
· Section 4.8: the authors should add to this section which software they utilised for statistical analysis.
Author Response
Comment 1:
Figure 3: please improve quality. The lines look blurred and the legend in the graph at the bottom left has been cut.
Response:
We have improved the quality of figure. We have confirmed that the lines are drawn clearly and all legend in graph are described correctly.
Comment 2:
Section 4.4: could the authors describe the method? The authors mentioned they treated the samples with Tewitro, what is Tewitro?
Response:
We have added the explanation of Tewitro TW24 which is the RHEK model-specific TEWL measuring device in the Material and Method section as follows (Page 11, Lines 330-338).
“Measurements were conducted before and 1, 3, 5, and 7 days after the treatment of samples with Tewitro TW24 (Courage+Khazaka, Cologne, Germany) which is the device specifically developed for the TEWL measurement on in vitro test and can directly and simultaneously measure the TEWL of RHEK model in a 24-well plate. The RHEK model was placed on a thermal insulation mat (HIENAI Mat; Cosmo Bio Co., Ltd., Tokyo, Japan), which maintained the entire bottom surface at 32℃, without a lid for 5 min before measurements. TEWL was measured for 30 min by inserting the sensor of Tewitro TW24 into each well of the RHEK model while maintaining the bottom surface at 32℃, and the mean value for the last 10 min was used in the analysis.”
Comment 3:
Section 4.7: please describe in depth electron microscopy method. SEM or TEM? Which microscope was used? Which parameters were used (working distance, accelerating voltage)? How where the samples prepared?
Response:
We have added the description about electron microscopy method in Materials and Methods section as follows (Page 12, Lines 374-379).
“The samples of tissues for TEM were fixed in phosphate buffered 2% glutaraldehyde, and subsequently post-fixed in 2% osmium tetra-oxide for 2 h in the ice bath. Then, the specimens were dehydrated in a graded ethanol and embedded in the epoxy resin. Ultrathin sections were obtained by ultramicrotome technique. Ultrathin sections stained with uranyl acetate for 15 min and lead staining solution for 2 min were submitted to TEM observation at 100 kV (H-7600, HITACHI Ltd., Tokyo, Japan).”
Comment 4:
Section 4.8: the authors should add to this section which software they utilised for statistical analysis.
Response:
We have added the description about the software which we used for statistical analysis as follows (Page 12, Line 381).
“All statistical analyses were performed using Statcel (OMS Ltd., Tokyo, Japan) as an add-in to excel.”